

# Three-dimensional analysis of facial morphology in nine-year-old children with different unilateral orofacial clefts compared to normative data

Marjolein Crins-de Koning[1], Robin Bruggink[1,2], Marloes Nienhuijs[3,4], Till Wagner[4,5], Ewald M. Bronkhorst[6] and Edwin M. Ongkosuwito[1,4]

[1] Section of Orthodontics and Craniofacial Biology, Department of Dentistry, Radboud University Medical Center, Nijmegen, Netherlands
[2] Radboudumc 3D Lab, Radboud Institute for Health Sciences, Radboud University Medical Center, Nijmegen, Netherlands
[3] Department of Oral and Maxillofacial Surgery, Radboud University Medical Center, Nijmegen, Netherlands
[4] Amalia Cleft and Craniofacial Center, Radboud University Medical Center, Nijmegen, Netherlands
[5] Department of Plastic and Reconstructive Surgery, Radboud University Medical Center, Nijmegen, Netherlands
[6] Department of Dentistry, Radboud University Medical Center, Nijmegen, Netherlands

Corresponding author
Marjolein Crins-de Koning,
marjolein.dekoning@radboudumc.nl

## ABSTRACT

**Aim:** To compare three-dimensional (3D) facial morphology of various unilateral cleft subphenotypes at 9-years of age to normative data using a general face template and automatic landmarking. The secondary objective is to compare facial morphology of 9-year-old children with unilateral fusion to differentiation defects.

**Methods:** 3D facial stereophotogrammetric images of 9-year-old unilateral cleft patients were imported into 3DMedX® for processing. All images of patients with a right sided cleft were mirrored. A regionalized general facial template was used for standardization. This template was pre-aligned to each face using five automatically determined landmarks and fitted using MeshMonk. All cleft patients were compared to an age-and gender matched normative face using distance maps and inter-surface distances (mm). Average faces were created for five groups (unilateral cleft lip, alveolus, and/or palate (UCL/A/P), fusion and differentiation defects). The selected regions for the evaluation of facial morphology were: complete face, nose, upper lip, lower lip, chin, forehead, and cheeks.

**Results:** A total of 86 consecutive 3D-stereophotogrammetry images were acquired for examination. No statistically significant differences were observed among the UCL, UCLA, and UCLP groups for the complete face, cheeks, chin, forehead, lower lip, and nose. However, in the upper lip region a significant difference was observed between the UCLP and UCL groups ($P = 0.004$, CI [$-2.93$ to $-0.48$]). Further visual examination of the distance maps indicated that more severe clefts corresponded to increased retrusion in the midface and the tip of the nose, though these differences were not statistically significant across groups. For fusion *vs* differentiation defects, no statistically significant differences were observed, neither for the complete face nor for any of the individual regions.
**Conclusion:** The findings demonstrate statistically significant differences in the upper lip region between children with UCL and those with UCLP, particularly with greater upper lip retrusion in the UCLP group. The use of color-coded distance maps revealed local variations and a trend of asymmetry in the nasal region, with increasing retrusion of the nose tip, upper lip, and cheeks correlating with the severity of the cleft. Although these trends were not statistically significant, they suggest a progressive facial retrusion pattern as cleft severity increases. For the secondary objective, no statistical differences were found between the facial morphology of children with fusion and differentiation defects, although a similar progression of maxillary retrusion was observed in the distance maps.

## INTRODUCTION

Orofacial clefts (OCs) are among the most common craniofacial birth defects and result from disruptions during embryonic development. These clefts appear when incomplete fusion of facial structures occurs, which can lead to various physical, functional, and aesthetic problems (*Salari et al., 2022*; *Jensen et al., 2023*). During the formation of the primary and secondary palate complex embryological processes take place, like outgrowth, fusion, and differentiation of the facial swellings and palatine processes (*Krapels et al., 2006*). Disturbance of these developmental processes which take place early in pregnancy can result in many different cleft types with varying degrees of severity.

The specific timing of this embryonic disruption determines the type and the severity of the cleft. Thus, variability in timing and underlying fusion and differentiation defects in embryogenesis lead to specific cleft subphenotypes, with an extensive variety in severity (*Vermeij-Keers et al., 2018*). For example, fusion defects of the primary palate occur earlier in the embryogenesis around week 4–7 post conception and are therefore more extensive than differentiation defects which occur around 7–12 weeks post conception.

This developmental variability has given rise to different classification systems that help clinicians and researchers categorize clefts more accurately (*Martin & Swan, 2023*). The Dutch registry by the Dutch association of Cleft and Craniofacial malformations (NVSCA) and subsequent classification was introduced in 2014 and is based on the human embryological development of the primary and secondary palate (*Luijsterburg, Rozendaal & Vermeij-Keers, 2014*; *Vermeij-Keers et al., 2018*). In this classification, clefts are first classified in three categories: cleft lip/alveolus, cleft lip/alveolus and palate, and cleft palate. Subsequently, each defect is classified as a fusion and/or differentiation defect related to timing in weeks in embryonic development. Table 1 provides the classification of the individual clefts according to *Vermeij-Keers et al. (2018)*. Classification systems could lead to better specified phenotypic groups in research, possibly leading to better scientific outcomes and ultimately leading to more optimal treatment outcomes.

**Table 1 Classification of the individual cleft anomalies of the primary and secondary palate according to timing and underlying fusion and differentiation mechanisms in embryogenesis.**

| Early embryonic period (4–7 weeks postconception) | Late embryonic period (7–12 weeks postconception) | |
|---|---|---|
| **Primary palate** | **Primary palate** | |
| **Fusion defects** | **Differentiation defects** | |
| – Complete CL | – Incomplete CL | |
| – Complete CA (extending to the incisive foramen) | – Submucous CL[b] | |
| – Incomplete CA (if the lip is normal or has a complete cleft) | – Hypoplastic lip | |
| | – Incomplete CA (if the lip has an incomplete/submucous cleft) | |
| | – Submucous CA | |
| | – Hypoplastic alveolus | |
| | **Secondary palate** | **Secondary palate** |
| | **Fusion defects** | **Differentiation defects** |
| | – Complete hard CP | – Submucous hard CP |
| | – Incomplete hard CP | – Hypoplastic hard palate |
| | – Complete soft CP | – Submucous soft CP (including uvula) |
| | – Incomplete soft CP | – Hypoplastic soft palate (including uvula) |
| | – Complete CU | |
| | – Incomplete CU | |

Notes:
CA, cleft alveolus; CL, cleft lip; CP, cleft palate; CU, cleft uvula.
[b] Congenital scar, forme fruste, and subsurface, subcutaneous, or microform cleft lip (Copied from *Vermeij-Keers et al., 2018*).

To register the treatment outcome and to longitudinally follow OC patients, the focus is slowly shifting from 2D lateral cephalograms using ionizing radiation to non-ionizing 3D stereophotogrammetry (*Brons et al., 2012*). However, reliable 3D photogrammetry devices can be expensive and the analysis to compare 3D stereo photograms is complex but quickly developing (*Brons et al., 2019*). To directly compare the facial morphology of OC-patients a regionalized general face template can be used (*Bruggink et al., 2022*). This template is pre-aligned to the 3D image based on several facial landmarks. However, manual landmarking is time-consuming and prone to error. To improve the alignment, a fully automatic landmarking process, based on deep learning, has recently been developed. This facilitates quantitative analysis of large 3D datasets and making it more reliable and repeatable (*Berends et al., 2024*).

Significant natural variability exists in human facial shape and morphology. Adequate research of facial morphology necessitates the utilization of extensive datasets comprising 3D images to characterize average facial structures alongside their inherent variations. Acquiring an adequate quantity of normative data by assembling a sizable cohort of young individuals can, however, be challenging. Predefined databases can be a solution for this problem. The '3D growth curves' study conducted by *Matthews et al. (2021)* has examined approximately 5,500 facial 3D images, providing a basis for the generation of simulated 3D facial templates corresponding to specific age groups and genders.

3D stereophotogrammetry has been used to evaluate facial growth and treatment outcomes in infants and younger children with OCs (*Kluge et al., 2023*; *Brons et al., 2019*; *Al-Rudainy et al., 2019*). After the lip surgery at 6 months the symmetry in the upper lip of

UCLP patients was restored, and the shape of the upper lip showed less variation after primary lip and soft palate closure. Even at this early age, retrusion of the soft-tissue mandible and chin was developing (*Brons et al., 2019*). Other studies have also investigated facial morphology and nasal soft tissue symmetry in older children with unilateral orofacial clefts and compared the results with non-cleft individuals (*Bugaighis et al., 2014*; *Kuijpers et al., 2021*). *Bugaighis et al. (2014)*, conducted a morphometric study with 3D facial morphological differences between average faces of 8- to 12-year old cleft children using landmark measurements. Significant differences were seen between the cleft groups where the lip is affected and the isolated cleft palate and control groups, mainly in the nasolabial region (*Bugaighis et al., 2014*). Furthermore, a recent study performed by *Kuijpers et al. (2021)* on the nasolabial shape and aesthetics in unilateral cleft patients using a mean 3D facial template showed shape differences between cleft faces and the average non-cleft face in children of 9- to 11-years old. The shape differences mainly affected the combined nasolabial area (*Kuijpers et al., 2021*). But none of these studies used automatic landmarking and a regionalized general face template as a reliable method for comparison. Furthermore, none of these studies compared different unilateral subphenotypes at 9 years of age to a normative population for the whole face and different regions. This is important because accurately quantifying craniofacial morphology, through comparison of cleft patients to normative values, can offer valuable insight into underlying pathological processes and forms a critical basis for treatment planning, and research (*Brons et al., 2012*).

The age of nine presents a compelling opportunity for the examination of facial morphology, as it precedes the onset of the pubertal growth spurt, during which facial aesthetics progressively gain significance for patients. Additionally, this coincides with the timing of the alveolar bone grafting procedure, which is commonly preceded by the acquisition of a 3D facial image.

The aim of this study is to compare 3D facial morphology of different unilateral cleft subphenotypes at 9-years of age to normative data using a general face template and automatic landmarking. The secondary objective is to compare facial morphology of 9-year-old children with unilateral fusion to differentiation defects.

# MATERIALS AND METHODS

## Patients

The study protocol received approval from the medical ethics commission of the Radboud University Medical Center (Commissie Mensgebonden Onderzoek 2021-13168).

The inclusion criteria were: (1) Diagnosis of a partial or complete unilateral orofacial cleft (unilateral cleft lip (UCL)/unilateral cleft lip alveolus (UCLA)/unilateral cleft lip and palate (UCLP)), (2) availability of a 3D stereophotogrammetry image of the face at the age of 9 years meeting predefined quality standards, defined as: absence of gaps or voids in the facial mesh, absence of hair on the face, neutral facial expression, and uniform mesh integrity, (3) documented and signed informed consent by parents or guardians, (4) both parents being of Caucasian descent, and (5) treatment administered within a unified cleft

team adhering to standardized institutional protocols. Exclusion criteria included the presence of syndromes or congenital malformations other than unilateral orofacial clefts.

The OC patients included in this study, received treatment by a unified multidisciplinary team at the Cleft Palate Craniofacial Unit of the Amalia Childrens Hospital, Radboud University Medical Center, including experienced surgeons. The surgery protocol employed herein encompasses primary lip surgery and soft palate closure at approximately 6 to 9 months of age, followed by hard palate closure at around 3 years of age. Prior to the eruption of the permanent canines, alveolar bone graft (ABG) surgery is performed, usually at around 9-years of age. The 3D images utilized in this study were captured prior to the ABG procedure, which aligns with the standard imaging protocol within our care regimen.

### 3D stereophotogrammetry images

3D images of cleft patients were acquired from the database of the Amalia Cleft Palate and Craniofacial Unit of the Radboudumc, Nijmegen, the Netherlands. These images were captured utilizing the 3dMD face™ System, 3dMD LLC, Atlanta, Georgia, USA under standardized conditions between 2016 and 2022. The age-and gender-matched normative faces were created using the open-source 3D growth curves database established by *Matthews et al. (2021)*.

### Processing

The data was organized into three experimental groups: UCL, UCLA, UCLP. Additionally, all anomalies were categorized as fusion or differentiation defects according to the classification system presented by *Vermeij-Keers et al. (2018)* as can be seen in Table 1. In cases where a Simonart's band was present ($n = 18$), it was classified as a differentiation defect.

Subsequently, all images were imported into 3DMedX® (v1.2.34.0; 3D Lab Radboudumc, Nijmegen, The Netherlands) for processing. To ensure uniformity within the dataset, images of patients with a right-sided cleft were mirrored, allowing better comparison. Standardization of the images involved the utilization of a regionalized general face template to indicate 7,160 quasi-landmarks on the face. Regional comparison was performed by dividing the template into distinct areas, including the total facial surface, nose, upper lip, lower lip, chin, forehead, and cheeks. The template was pre-aligned to each face using five automatically identified landmarks (Figs. 1.1 and 1.2) before the fitting process within the MeshMonk toolkit using first a rigid Procrustes algorithm and afterwards a non-rigid registration for further alignment (*White et al., 2019*). The automatic landmarking, based on artificial intelligence, is described by *Berends et al. (2024)*.

All cleft patients were compared to an age- and gender-matched normative face using distance maps, as shown in Figs. 1.3 and 1.4. The cleft template and the normative template (Figs. 1.2 and 1.3) were aligned using a Robust superimposition algorithm (*Claes, Walters & Clement, 2012*). Inter-surface distance, measured in millimeters, was employed to indicate differences. The inter-surface distances were grouped into regions to asses

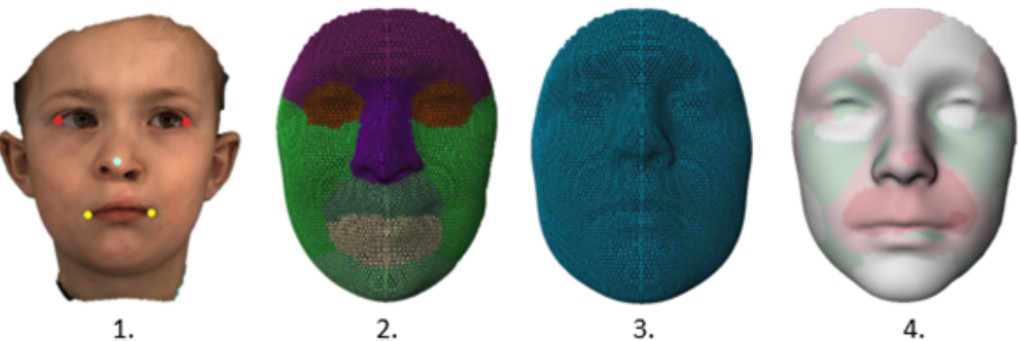

**Figure 1 Overview of the mesh processing in four steps:.** 1. The 3D image featuring five automatically identified landmarks (Exocanthions, Nasal tip, and Cheilions) 2. Fitting of the regionalized general face template onto the 3D image. 3. Alignment of the expected normative face template matched for age and gender 4. Generation of the distance map illustrating the superimposition of step 2 and 3 using a robust superimposition algorithm.

regional alterations between cleft patients and the normative control model. A more comprehensive description of the method can be found in the article of *Bruggink et al. (2022)*.

## Statistical analysis

No inter- or intraobserver reliability assessments were conducted because of the fully automated nature of the process, which eliminated human error. The automatic landmarking has been shown to be comparable to manual annotation, with an excellent mean precision of $1.69 \pm 1.15$ mm and an inter-observer variability of $1.31 \pm 0.91$ mm. Additionally, the reliability of the MeshMonk method averaged $1.97 \pm 1.34$ mm (*Berends et al., 2024*).

T-test were performed to compare the cleft children to the normative control model. This was done for the whole face and for specified regions. Additionally, to compare the unilateral cleft groups, an ANOVA and Tukey *post hoc* tests were performed. Fusion and differentiation defect analyses were conducted using a t-test. The significance level was set at $P \leq 0.05$.

## RESULTS

A total of 86 3D-stereophotogrammetry images of 9-year-old cleft patients, born between 2007 and 2013, were acquired. Among them, 25 patients presented with UCL (mean age: 9 y 2 m, 19 boys, six girls), 21 with UCLA (mean age: 9 y 3 m, nine boys, 12 girls) and 40 with UCLP (mean age: 9 y 3 m, 27 boys, 13 girls). Of these 86 cases, 52 presented a differentiation defect (mean age: 9 y 2 m, 35 boys, 17 girls) and 34 exhibited a fusion defect (mean age: 9 y 3 m, 20 boys, 14 girls).

Table 2 provides an overview of the included 3D images and their respective characteristics. Among the 46 excluded images, 23 were excluded due to non-Caucasian descent, 12 were syndromic clefts, five were isolated cleft palates, and six were excluded due to a non-resting facial expression.

**Table 2 The number of included 3D images and their characteristics.**

| Group | N | UCL | UCLA | UCLP |
|---|---|---|---|---|
| 3D images in database (unilateral OCs, 9-year olds) | 132 | | | |
| Excluded images | 46 | | | |
| Total meshes | 86 | 25 | 21 | 40 |
| Differentiation defects | 52 | 25 | 14 | 13 |
| Fusion defects | 34 | 0 | 7 | 27 |
| Boys | 55 | 19 | 9 | 27 |
| Girls | 31 | 6 | 12 | 13 |

**Table 3 Comparison of mean difference between groups and controls in millimeters.**

| Region | UCL (N = 25) | UCLA (N = 21) | UCLP (N = 40) | Differentiation (N = 52) | Fusion (N = 34) |
|---|---|---|---|---|---|
| | Mean difference in millimeters, (*P* value), [95% Confidence interval] | | | | |
| All | −0.06 (0.874) | −0.51 (0.214) | −0.34 (0.37) | −0.24 (0.375) | −0.40 (0.35) |
| | [−0.89 to 0.77] | [−1.33 to 0.32] | [−1.11 to 0.42] | [−0.76 to 0.29] | [−1.27 to 0.46] |
| Cheeks | −0.43 (0.432) | −1.17 (0.041) | −0.82 (0.089) | −0.66 (0.062) | −1.00 (0.073) |
| | [−1.53 to 0.67] | [−2.30 to −0.05] | [−1.78 to 0.13] | [−1.35 to 0.04] | [−2.11 to 0.10] |
| Chin | 0.48 (0.446) | 0.38 (0.56) | 0.67 (0.255) | 0.43 (0.316) | 0.72 (0.264) |
| | [−0.80 to 1.76] | [−0.95 to 1.70] | [−0.50 to 1.85] | [−0.42 to 1.28] | [−0.57 to 2.01] |
| Forehead | 0.45 (0.374) | 0.23 (0.643) | −0.13 (0.754) | 0.20 (0.553) | 0.03 (0.954) |
| | [−0.58 to 1.48] | [−0.79 to 1.25] | [−0.94 to 0.69] | [−0.46 to 0.85] | [−0.89 to 0.94] |
| Lower lip | −0.02 (0.961) | −0.40 (0.272) | 0.31 (0.323) | 0.02 (0.945) | 0.08 (0.814) |
| | [−0.76 to 0.73] | [−1.13 to 0.34] | [−0.32 to 0.94] | [−0.46 to 0.49] | [−0.63 to 0.80] |
| Nose | −0.10 (0.77) | −0.08 (0.774) | −0.11 (0.695) | −0.06 (0.77) | −0.17 (0.619) |
| | [−0.79 to 0.59] | [−0.66 to 0.50] | [−0.70 to 0.47] | [−0.46 to 0.34] | [−0.85 to 0.51] |
| Upper lip | 0.20 (0.601) | −0.55 (0.201) | −1.50 (<0.001) | −0.40 (0.123) | −1.34 (0.003) |
| | [−0.59 to 0.99] | [−1.41 to 0.31] | [−2.19 to −0.82] | [−0.91 to 0.11] | [−2.19 to −0.50] |

Comparison for the mean distances between unilateral cleft groups and their matched controls, categorized per group and facial region, are provided in Table 3. The cheeks of UCLA children differed significantly from the norm ($P = 0.041$, CI [−2.30 to −0.05]). The UCLP patients differed significantly from the norm in the upper lip region ($P < 0.001$, CI [−2.19 to −0.82]). The statistical comparison between the groups is shown in Table 4. No statistically significant differences were observed among the UCL, UCLA, and UCLP groups for the complete face, cheeks, chin, forehead, lower lip, and nose. However, significant differences were identified among the groups for the upper lip region ($P = 0.005$). Specifically, the UCL group showed a mean distance of 0.20 mm, the UCLA group −0.55 mm, and the UCLP group −1.50 mm. Subsequent Tukey *post hoc* test revealed a significant difference only between the UCLP and UCL groups ($P = 0.004$, CI [−2.93 to −0.48]), as illustrated in Table 5. Notably, the mean distance for the UCLA upper lip did

**Table 4 P-values of the ANOVA comparison between the UCL/UCLA/UCLP groups, and the statistical comparison of fusion vs. differentiation defects (t-tests).**

| Region | UCL/UCLA/UCLP Anova | Fusion vs. differentiation T-tests |
|---|---|---|
| All | 0.777 | 0.738 |
| Cheeks | 0.661 | 0.591 |
| Chin | 0.941 | 0.702 |
| Forehead | 0.640 | 0.762 |
| Lower lip | 0.358 | 0.874 |
| Nose | 0.997 | 0.780 |
| Upper lip | 0.005* | 0.057 |

Note:
* Statistically significant difference.

**Table 5 Tukey post hoc multiple comparisons of means (in mm) of the upper lip region of the three cleft groups.**

| Comparing groups | Mean difference (mm) | Confidence interval 95% (mm) | P value |
|---|---|---|---|
| UCLA-UCL | −0.75 | [−2.17 to 0.68] | 0.425 |
| UCLP-UCL | −1.70 | [−2.93 to −0.48] | 0.004* |
| UCLP-UCLA | −0.96 | [−2.25 to 0.34] | 0.189 |

Note:
* Statistically significant difference.

not significantly differ from either the UCL or UCLP group. Figure 2 shows these differences in color-coded distance maps.

Further visual examination of the distance maps indicated that more severe clefts corresponded to increased retrusion in the midface and the tip of the nose, though these differences were not statistically significant across groups.

For the second objective of this study, a comparison was made between fusion and differentiation defects. The mean 3D distances between cleft patients and their normative matched controls are presented in Table 3 for both the complete face and its separate regions. In Table 4, the comparison between fusion vs differentiation defects is presented. Only the upper lip region of fusion defect patients differed statistically from the norm ($P = 0.003$, CI [−2.19 to −0.50]). When comparing fusion and differentiation defects, no statistically significant differences were observed, neither for the complete face nor for any of the individual regions. Visual representations of these differences are depicted in the color-coded distance maps presented in Fig. 2.

## DISCUSSION

The aim of this study was to assess the 3D facial morphology of various unilateral cleft subphenotypes at 9-years of age compared to normative data, using a general face template and automatic landmarking. The findings revealed statistically significant differences in the upper lip region between the UCL and UCLP group, with the UCLP group having greater overall retrusion of the upper lip. Local differences could be visualized through color-coded distance maps. These maps provided a trend illustrating asymmetry in the

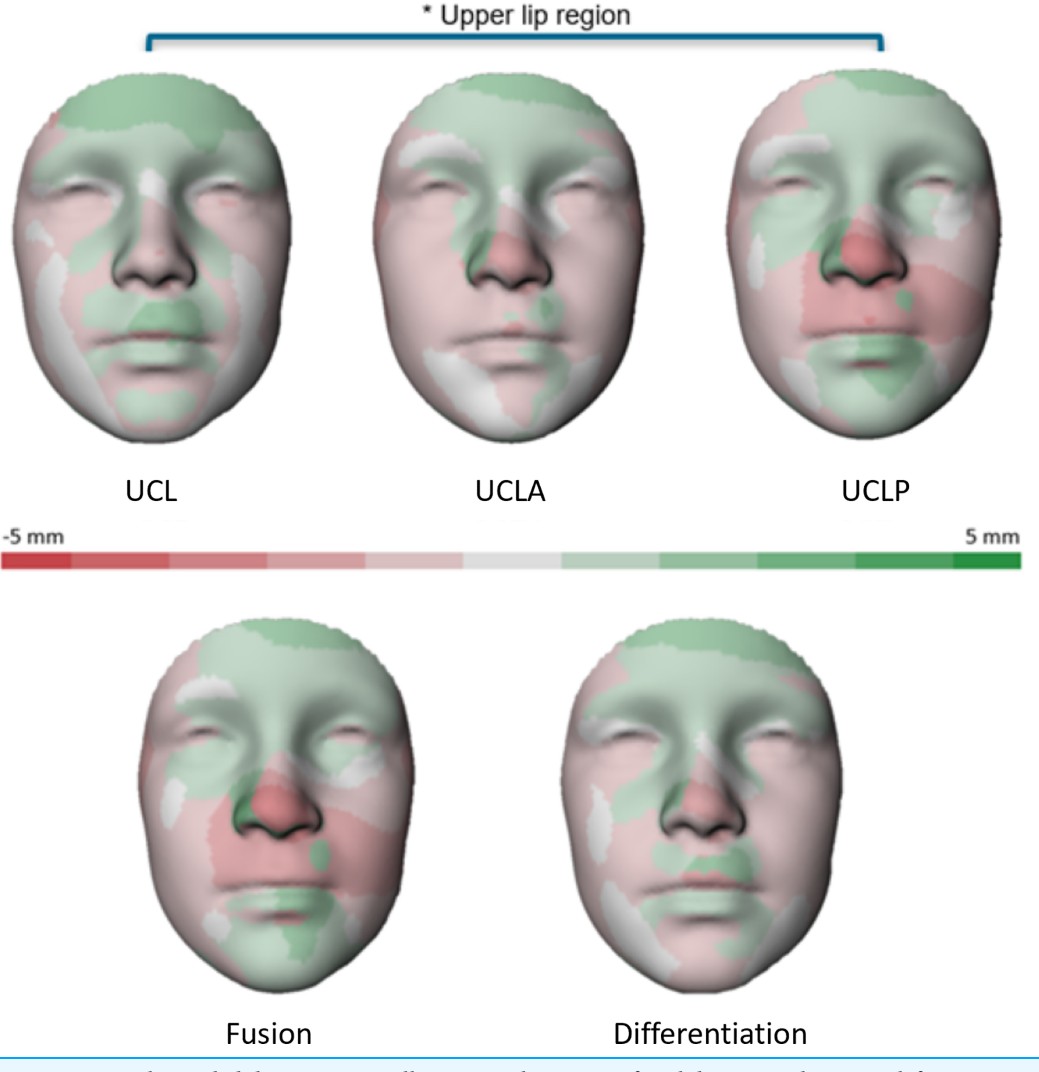

**Figure 2 Color-coded distance maps illustrating the average facial disparities between cleft patients and their matched controls across various groups (UCL/UCLA/UCLP/Fusion/Differentiation).** An asterisk (*) shows statistically significant differences between UCL and UCLP in the upper lip region (*P* = 0.004).

nose and a gradual progression of the retrusion in the nose tip, upper lip, and cheeks as cleft severity increased, although these observations lacked statistical significance. However, the distribution of the fusion and differentiation cases within the cleft groups is noteworthy (see Table 2). For example, all 25 UCL patients were differentiation defects (incomplete UCL or submucous UCL or had a Simonart's bands) in this sample. The unequal distribution of fusion and differentiation defects across the groups could affect the overall results, particularly in terms of the subtle morphological differences observed. Furthermore, the sample sizes for each subgroup may limit the study's statistical power, making it more challenging to detect significant differences.

For the secondary objective, fusion and differentiation defects were compared, yielding no statistical differences. Nonetheless, a similar trend of progressing retrusion of the maxillary area increasing with cleft severity was observed in the distance maps. These

findings highlight the complexity of comparing facial morphology, yet the combination of the data and the distance maps provide a comprehensive overview of the facial morphology of OC patients at 9 years of age.

Even though the observed differences were subtle and only significant in the upper lip region, this holds potential clinical relevance because this study presents a fast, easy, and reliable way to asses facial morphology of individual patients compared to normative faces. This helps to further evaluate and improve surgical interventions and timing, achieving optimal aesthetic and functional outcomes. By integrating these subtle morphological differences into preoperative planning and long-term follow-up, clinicians may be better equipped to individualize treatment strategies, ultimately enhancing patient-specific care and improving overall satisfaction with facial appearance and function.

In contrast to our study, Bugaighis et al. (2014) reported numerous significant differences in their 3D comparison of facial morphology of subjects with UCLP or UCLA compared to average controls. These differences were predominantly asymmetry of the face that was concentrated in the nasolabial region. Unlike our study, Bugaighis et al. (2014) utilized point measurements instead of mean differences in individual facial regions. While our study also identified the labial area as notably different, consistent with Bugaighis et al.'s (2014) findings, statistically differences in the nose region were not observed. One possible explanation for this disparity could be attributed to the sample size in our study, potentially limiting the detection of subtle differences. Additionally, the regional analysis employed in our study provides an overall mean difference but this will level out small areas of the face, for example the scar area in the upper lip and the asymmetry of the nostrils.

Despite the inherent differences in growth and development between fusion and differentiation clefts, we did not observe significant differences between these subtypes. Conversely, the surgical technique and experience of the surgeon plays a crucial role in obtaining an optimal facial morphology (Fell et al., 2024). In cases of closing a partial cleft lip with a differentiation origin, the initial surgical step involves transforming the partial cleft into a complete one. This process is necessary for the reconstruction and preparation of the muscle, which must be detached from its incorrect attachment to the base of the skull and repositioned between the skin and mucosa. However, the initial condition with an incomplete cleft presents a distinct scenario, characterized by less asymmetry in the nostrils.

In contemporary practice, 3D stereophotogrammetry has become increasingly prevalent as a method for documenting and studying facial morphology. The rapidly evolving analysis tools enables qualitative assessment of large datasets, with artificial intelligence playing a significant role in reducing reproducibility errors. The automatic method utilized in this study has undergone rigorous testing, as detailed by Bruggink et al. (2022). Reliability analyses demonstrated high agreement between and within observers, with no significant differences observed (Bruggink et al., 2022). Small differences, however, could be observed. These could be attributed to manual landmarking. Berends et al. (2024), later improved the method and demonstrated no significant differences between the

automatic or manual landmarking. Notably, this method exhibits a mean precision of 1.69 ± 1.15 mm and an inter-observer variability of 1.31 ± 0.91 mm (*Berends et al., 2024*), rendering it well-suited for comparing and evaluating facial morphology. In the future, this may even be further improved by artificial intelligence.

However, a primary limitation of our study is the utilization of normative expected facial shape as a control group, which is derived from a diverse dataset of individuals primarily form the UK, Australia, and the USA with European genomic ancestry. Despite differences in facial shape between these populations and Dutch children, local control groups could not be employed to the age at which healthy patients in the Netherlands are typically referred for orthodontic care, typically around 10–12 years old, resulting in an insufficient number of available 3D images. For future research, it would be valuable to compare with a sizable local population control-group and, also, expand the cleft population to mitigate the impact of outliers.

We acknowledge that facial morphology is influenced by many factors such as ancestry, age, and gender, height, weight, BMI, and mandibular plane angle (*De Greef et al., 2006*; *Godt et al., 2013*). While our study primarily focused on ancestry, age, and gender as matching variables, we recognize that the omission of additional factors could introduce potential sample bias. Obtaining normative data matched for all these variables presents significant challenges due to the complexity and diversity of craniofacial characteristics. This highlights the need for further research that incorporates a broader range of variables.

## CONCLUSIONS

The findings demonstrate statistically significant differences in the upper lip region between children with UCL and those with UCLP, particularly with greater upper lip retrusion in the UCLP group. The use of color-coded distance maps revealed local variations and a trend of asymmetry in the nasal region, with increasing retrusion of the nose tip, upper lip, and cheeks correlating with the severity of the cleft. Although these trends were not statistically significant, they suggest a progressive facial retrusion pattern as cleft severity increases.

For the secondary objective, no statistical differences were found between the facial morphology of children with fusion and differentiation defects, although a similar progression of maxillary retrusion was observed in the distance maps.

Future research should explore larger cohorts and further refine the analysis of facial morphology and retrusion patterns to enhance our understanding of cleft phenotypes and their impact on facial development.

## ACKNOWLEDGEMENTS

We would like to express our gratitude to P. Claes and H. Matthews of the KU Leuven for publishing their large-scale open-source 3D growth curves and helping us to incorporate this in our study for the normative data.

Furthermore, we would like to acknowledge the use of the AI tool by *Berends et al. (2024)* for the automatic landmarking as described in the method. We also used Grammarly (2024) and OpenAI (2024) for style and grammar corrections.

### Funding

The authors received no funding for this work.

### Competing Interests

Robin Bruggink is a developer for 3DMedX® which is a commercial research package used in this study.

### Author Contributions

- Marjolein Crins-de Koning conceived and designed the experiments, performed the experiments, analyzed the data, prepared figures and/or tables, authored or reviewed drafts of the article, and approved the final draft.
- Robin Bruggink conceived and designed the experiments, performed the experiments, analyzed the data, prepared figures and/or tables, authored or reviewed drafts of the article, and approved the final draft.
- Marloes Nienhuijs analyzed the data, authored or reviewed drafts of the article, and approved the final draft.
- Till Wagner analyzed the data, authored or reviewed drafts of the article, and approved the final draft.
- Ewald M. Bronkhorst analyzed the data, prepared figures and/or tables, authored or reviewed drafts of the article, and approved the final draft.
- Edwin M. Ongkosuwito conceived and designed the experiments, analyzed the data, authored or reviewed drafts of the article, and approved the final draft.

### Human Ethics

The following information was supplied relating to ethical approvals (*i.e.*, approving body and any reference numbers):

The study protocol received approval from the Radboud University Medical Center medical ethics committee (CMO 2021-13168).

### Data Availability

The raw measurements are available in the Supplemental File.

### Supplemental Information

Supplemental information for this article can be found online at http://dx.doi.org/10.7717/peerj.18739#supplemental-information.

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
