# Peer review of "Three-dimensional analysis of facial morphology in nine-year-old children with different unilateral orofacial clefts compared to normative data"

_PeerJ, doi:10.7717/peerj.18739_

## Round 0.1 · original submission · Major Revisions

Although the study is interesting, the reviewers have highlighted several issues that need to be addressed before further consideration. Additionally, I would like to draw your attention to the impact of the different distributions of fusion and differentiation cases within the cleft groups (as shown in Table 1) on the outcomes reported in Tables 2 and 3. The sample size of each group, along with various factors that influence the outcomes, and the method’s precision—while acceptable, but not excellent, as stated in the manuscript—may also contribute to the lack of statistically significant results. These issues should be discussed and clearly reported in the manuscript.

·

Basic reporting

In this study, de Koning et al. analyzed 86 3D stereophotogrammetric facial images of UOC patients and compared them with age- and gender-matched normative faces. Additionally, comparisons were made between subtypes of UOC (UCL, UCLA, and UCLP). The authors concluded that UCLA patients exhibited more retruded cheeks, while UCLP and fusion defect patients demonstrated a more collapsed upper lip in comparison to non-affected individuals. Statistically significant differences were also observed between UCL and UCLP in the upper lip region.

This work holds potential clinical significance for the assessment and treatment of OC patients. However, several issues need to be addressed. Therefore, I recommend MAJOR REVISION and outline the concerns below.

Experimental design

No comment

Validity of the findings

No comment

Additional comments

In this study, de Koning et al. analyzed 86 3D stereophotogrammetric facial images of UOC patients and compared them with age- and gender-matched normative faces. Additionally, comparisons were made between subtypes of UOC (UCL, UCLA, and UCLP). The authors concluded that UCLA patients exhibited more retruded cheeks, while UCLP and fusion defect patients demonstrated a more collapsed upper lip in comparison to non-affected individuals. Statistically significant differences were also observed between UCL and UCLP in the upper lip region.

This work holds potential clinical significance for the assessment and treatment of OC patients. However, several issues need to be addressed. Therefore, I recommend MAJOR REVISION and outline the concerns below.

(1) The clinical relevance of this study for OC treatment should be clearly stated. Has there been any controversy regarding the facial features of OC patients (compared to non-affected children) in previous studies? Why was it necessary to perform a superimposition analysis between OC patients and normative faces? The manuscript does not address these questions.

(2) Figure 2 seems to be the core data in this manuscript. However, I have a few concerns when interpreting it alongside the Results section. Why do UCL patients display a more protrusive upper lip than normative faces (as indicated by the green color in the upper lip)? Although cleft lip only (CLO) does not directly affect maxillary development, it would be expected, both theoretically and based on clinical experience, that the cleft side of the upper lip would not appear more protrusive. While I do not work in this exact area, I propose that the landmarks used to align the 3D stereophotogrammetric images may not sufficiently capture the facial features (from Figure 1.1 to 1.2). Could the authors add more landmarks relevant to lip morphology, such as peak points on the affected and non-affected sides? And the red underline in the lower panel of Figure 2 should be removed.

(3) One critical issue in this study is the potential sample bias. Facial morphology is influenced by numerous factors beyond age and gender (the authors have used in the current study), such as height, weight, BMI, and mandibular plane angle. Although obtaining normative faces matched for all of these variables is difficult, the authors should discuss the limitations regarding potential sample bias.

(4) Lines 238-244 should be moved to the Materials & Methods section.

Reviewer 2 ·

Basic reporting

The manuscript presents an interesting approach by categorizing the face into both the entire face and specific regions (nose, upper lip, lower lip, chin, forehead, and cheeks). However, this categorization could benefit from more clarification, as treating the entire face and individual regions as parallel categories is not entirely clear. While this approach could potentially offer valuable insights, a more detailed explanation of the distinction between the whole face template and specific region templates would help to better justify the choice and enhance the study's design.

Experimental design

The use of automated landmarking and a general facial template for 3D analysis is an interesting and innovative choice. However, it remains unclear if this method offers clear advantages over traditional techniques. While the findings regarding the upper lip are insightful, similar results may potentially be achieved through more conventional two-dimensional or manual methods. Moreover, the authors mention that the method does not allow for the assessment of facial symmetry, which is particularly important for evaluating unilateral cleft lip and palate patients. Since symmetry is a key indicator of successful treatment in these cases, this limitation raises some concern about whether the method is fully suited to the study’s goals. Providing more clarity on this limitation and its implications would give a better understanding of the method’s applicability.

Validity of the findings

The study identifies some subtle differences in the upper lip region between different cleft subtypes (UCL, UCLA, UCLP), which is noteworthy. However, it would be helpful if the manuscript could provide a more explicit discussion on the clinical relevance of these findings. Expanding on how these results might influence treatment planning or assessment of outcomes could strengthen the study’s impact and make it more applicable for clinical practice. This additional context would further clarify the significance of the observed differences.

Reviewer 3 ·

Basic reporting

In my opinion, the article is generally well written. However, in my opinion the introduction should be substantially revised. I suggest providing clear explanations instead of informing the reader what reference contains the knowledge essential to understand the manuscript. All the sentences should logically follow one another in order to bring the reader up to date knowledge on:
1. The etiology of clefts and the existence of different classifications of orofacial clefts.
The general sentences like: Variability in timing and underlying fusion and differentiation defects in embryogenesis lead to specific cleft subphenotypes, with an extensive variety in severity” or “ Fusion defects occur earlier in the embryogenesis and are therefore more extensive than differentiation defects. Because of this phenotypic variation, different classification systems exist for orofacial clefts.” could be expanded to provide knowledge and explanation to the readers (allowing to understand the paper without reading the cited studies).

The authors should clearly explain to the reader what is the visual difference between fusion and differentiation defects and what are the 9 subfenotypes.
The first five sentences of the introduction could be removed from the introduction, as they do not refer to craniofacial morphology.

2. Methods to assess 3D craniofacial morphology - this knowledge is already provided
3. Characteristics of craniofacial morphology of cleft patients – what is the main difference comparing to uncleft individuals according to the literature

I find the presentation of the results correct.
The discussion is concise and refers strictly to the subject of the study.

Experimental design

I find the title of the paper: “THREE – DIMENSIONAL FACIAL MORPHOLOGY OF UNILATERAL CLEFT SUBPHENOTYPES AT NINE YEARS OF AGE” inconsistent with the aim of the study. The aim is not to describe the 3D craniofacial morphology. It is “to compare 3D facial morphology of different unilateral cleft subphenotypes at 9-years of age to normative data using a general face template and automatic landmarking. The secondary objective is to compare facial morphology of 9-year-old children with unilateral fusion to differentiation defects.”
Thus, I suggest to modify the title according to the aim. Moreover, the term “cleft subphenotypes” may not be clear to the average reader and thus should not appear in the title.

I find the methods correct and adequate.
In the "material and methods" the authors should clearly explain what criteria were used to diagnose a fusion or a differentiation defect in this study. Besides this, I find the study design and the "material and methods" chapter acceptable.

Validity of the findings

This study is promising new knowledge on craniofacial morphology from 3D analysis, however, conclusions are not linked to the aims of the study.
Please, see the aim of the study and provide here response to the aims. Additional conclusions may appear at the end.

---

## Round 0.2 · Minor Revisions

The reviewers and I are satisfied with the revision, which has adequately addressed all concerns. However, I suggest simplifying the study title to: 'Three-dimensional analysis of facial morphology in nine-year-old children with different unilateral orofacial clefts compared to normative data.' Please make this revision and resubmit your manuscript for acceptance and publication.

·

Basic reporting

The authors have addressed all my concerns during this revision, and I have suggested that it be accepted.

Experimental design

No comment.

Validity of the findings

No comment.

Additional comments

No comment.

Reviewer 3 ·

Basic reporting

The authors have substantially revised their manuscript. All the remarks and suggestions by reviewers have been addressed.

Experimental design

The description of methods has been revised. Sufficient explanations have been provided, as recommended.

Validity of the findings

All relevant data have been adequately presented. The conclusions have been revised properly.

Additional comments

English requires tiny corrections - I suggest careful reading the manuscript.

---

## Round 0.3 · accepted · Accept

The revised manuscript adequately addresses all concerns raised by the reviewers and myself. As the requested changes in this revision round were minor, I conducted this assessment personally. The current version is ready for publication.